# Evaluation of economic strengthening in South Africa and its impact on HIV, sexually transmitted infections, and teenage births: A modelling study

Leigh F. Johnson [1]*, Lise Jamieson [2,3,4], Mmamapudi Kubjane [2], Gesine Meyer-Rath [2,4,5]

1 Centre for Integrated Data and Epidemiological Research, University of Cape Town, Cape Town, South Africa, 2 Health Economics and Epidemiology Research Office, University of Witwatersrand, Johannesburg, South Africa, 3 Department of Medical Microbiology, Amsterdam University Medical Centre, Amsterdam, Netherlands, 4 The South African Department of Science and Innovation/National Research Foundation Centre of Excellence in Epidemiological Modelling and Analysis (SACEMA), Stellenbosch University, Stellenbosch, South Africa, 5 Department of Global Health, Boston University, Boston, Massachusetts, United States of America

* Leigh.Johnson@uct.ac.za

## Abstract

### Background

High incidence rates of HIV, sexually transmitted infections (STIs), and teenage pregnancy are major challenges facing South Africa. The role of socio-economic factors in driving these incidence rates is complex, with high socio-economic status protecting against some risk behaviours (condomless sex, early sexual debut, and casual/transactional sex in females) but increasing other risk behaviours (e.g., male engagement in casual and commercial sex). We aimed to model the effect of socio-economic status, and associated economic strengthening interventions, in South Africa.

### Methods and findings

We extended a previously-developed agent-based model of HIV, STIs, and fertility in South Africa to assess effects of education, employment, and per capita household income on sexual behaviours. We estimated these effects from literature and from calibration of the model to African randomized controlled trials of economic strengthening interventions. Population attributable fractions (PAFs) were calculated. We considered three intervention types, all targeting households with log per capita income below the national average: school support to reduce school dropout; vocational training for unemployed adults; and unconditional cash transfers. We estimate that low socio-economic status accounted for 13% of new HIV infections, 7% of incident STIs (gonorrhoea, chlamydia, and trichomoniasis) and 31% of teenage births in South Africa, over 2000−2020. However, because of uncertainties regarding effect sizes, confidence intervals around these PAFs are wide (1,50% for HIV; −1,19% for

which permits unrestricted use, distribution, and reproduction in any medium, provided the original author and source are credited.

**Data availability statement:** This study relies only on publicly available data, which are referenced throughout the manuscript. The most important data, used in model calibration, are the summary estimates of intervention effectiveness from the various randomized controlled trials of economic strengthening interventions, which are shown in Table ZB of S1 Text. We have also made the C++ code available on a github repository: https://github.com/leighjohnson/MicroCOSM.

**Funding:** L.F.J. received funding from the Gates Foundation (awards 007145 and 064474): https://www.gatesfoundation.org/. The funder had no role in the study design, analysis, decision to publish, or manuscript preparation.

**Competing interests:** The authors have declared that no competing interests exist.

**Abbreviations:** APCHI, adjusted per capita household income; ART, antiretroviral treatment; OR, odds ratio; PAFs, population attributable fractions; PRCCs, partial rank correlation coefficients; RCTs, randomized controlled trials; STIs, sexually transmitted infections; VMMC, voluntary medical male circumcision.

STIs; and 10,76% for teenage births). Over 2025−2040, none of the interventions are estimated to reduce HIV, STIs, or teenage births significantly, due to limited impact on secondary economic outcomes. The greatest impact would be that of school support on teenage births (a 5% reduction, 95% CI: −1,12%). Key limitations include the assumption of uniform STI treatment access across socio-economic strata, and the exclusion of possible socio-economic effects at a community level.

## Conclusions

Although poverty is likely to be a significant driver of HIV, STIs, and teenage pregnancy in South Africa, precise quantification is challenging. Recently trialled economic strengthening interventions have insufficient impact on socio-economic status to reduce HIV and STIs significantly at a population level.

---

### Author summary
#### Why was this study done?

- There is a high incidence of HIV, sexually transmitted infections (STIs), and teenage pregnancy in many sub-Saharan African countries, which is often blamed on poverty.

- However, evidence of the relationship between socio-economic status and HIV and STIs is inconsistent, with studies suggesting that the relationship varies over time and between settings.

- Randomized controlled trials (RCTs) of economic strengthening interventions have been similarly inconsistent regarding the impact of these interventions on HIV, STIs, and teenage pregnancy. It remains unclear which economic strengthening interventions should be prioritised.

- South Africa has the largest HIV epidemic in the world as well as the highest index of income inequality, and is therefore an important setting in which to assess the potential impact of economic strengthening.

#### What did the researchers do and find?

- We developed a mathematical model of the South African population, simulating changes over time in education, employment, household income, sexual behaviour, HIV, STIs, and teenage births. We simulated three economic strengthening interventions: school support to prevent school dropout, vocational training to increase employment, and cash transfers (income support for low-income households).

- The model estimates that over the 2000–2020 period low socio-economic status accounted for 13% of new HIV infections, 7% of new STIs and 31% of teenage

births in South Africa. However, because of uncertainty in many of the socio-economic effects, the 95% confidence intervals around these estimates were very wide.

- The model estimates that the economic strengthening interventions would have only modest effects on socio-economic indicators; for example, vocational training would only increase employment rates by 1% over the 2025–2040 period.

- Because these socio-economic impacts are small, none of the interventions are predicted to significantly reduce the incidence of HIV, STIs, or teenage births by 2040.

## What do these findings mean?

- Although poverty has contributed to the high rates of HIV, STIs, and teenage pregnancy in South Africa, recently trialled economic strengthening interventions are unlikely to have much impact on these outcomes over the next 15 years.

- In a context of extreme income inequality, more radical economic interventions would be needed to reduce HIV and STI incidence. Our findings may be less applicable in other African settings, where there is typically more poverty but lower income inequality.

- Despite the disappointing results, there may still be a role for economic strengthening interventions. Future modelling studies should also evaluate their impact on outcomes such as mental health, tuberculosis, maternal and child health, and social well-being.

- A limitation is that some socio-economic effects were not represented in the model, e.g., effects of socio-economic status on access to STI treatment, and effects of community socio-economic status.

## Introduction

Countries in southern and eastern Africa have the highest levels of HIV prevalence globally [1], as well as high incidence of curable sexually transmitted infections (STIs) [2] and teenage fertility [3]. High levels of poverty in the region are often blamed for these reproductive health challenges [4,5]. Teenage pregnancy in Africa is indeed associated with poverty, low female employment, and incomplete schooling [3,6–9]. However, the relationship between socio-economic status and the risk of HIV and STIs in Africa is more complex. For example, early reviews documented a *positive* relationship between socio-economic status and HIV in Africa [10–12], but later reviews have found a more nuanced picture, with suggestions of a change in the relationship between HIV and socio-economic status over time [13–16] and variation in the relationship across regions within Africa [15–21]. Similarly, STIs are often associated with employment [22,23] and wealth [23], but have also been found to be associated with low educational attainment [23,24].

The complexity of the relationship between socio-economic status and HIV/STI risk reflects a diversity of ways in which education, employment, and wealth can both mitigate and potentiate sexual risk behaviour. Higher educational attainment is strongly associated with greater condom use in nonspousal relationships [11,25–28], and schooling and household income both delay the timing of female sexual debut and entry into marriage [11,27,29–33]. In men, employment may be associated with engaging in commercial and transactional sex [34–36] as well as multiple partnerships [31,37]. In women, on the other hand, lower income is associated with transactional sex [38–40]. Factors other than sexual behaviour may also affect the observed association between HIV/STIs and socio-economic status. For example, voluntary medical male circumcision (VMMC), which protects against male HIV and STI acquisition, is more common in men with greater wealth [41,42]. A higher HIV prevalence in higher socio-economic strata might be a reflection of longer survival [43], due to higher

rates of HIV testing [16,44] and better access to treatment in wealthier individuals. The relationship between wealth and HIV might also be partly confounded by urbanicity, with urban areas having both higher levels of wealth and higher HIV prevalence [14,18].

Economic strengthening interventions have been developed in an attempt to address some of the socio-economic drivers of HIV, STIs, and teenage pregnancy. The most commonly tested interventions include cash transfers (either unconditional or conditional upon achieving certain educational/health outcomes), educational support, vocational training, microcredit schemes, and food assistance, among others [45]. Although reviews of the impacts of these interventions have noted some reductions in self-reported risk behaviours and improved contraceptive knowledge, there is relatively little evidence of changes in clinical outcomes (HIV and STI incidence and unintended pregnancy) [45–49]. Very few studies evaluated whether intervention impacts were sustained over the longer-term [46] or whether the interventions themselves were sustainable [49]. Data from randomized controlled trials (RCTs) of these interventions have not been systematically synthesised using traditional meta-analytic techniques, a reflection of the heterogeneity in interventions and trial outcomes. There is a lack of consensus on which interventions are most appropriate and in what contexts.

Mathematical models are widely used to inform policy decisions around HIV and reproductive health [50–52], and play a particularly important role in evaluating the relative cost-effectiveness of different interventions and the optimal allocation of scarce healthcare resources. They also play a role in translating evidence from RCTs, which produce short-term estimates of impact at an individual or community level, into estimates of likely longer-term impact at a population level. Yet there is a notable lack of mathematical modelling of economic strengthening interventions [53]. This is partly a reflection of the challenges associated with representing the complex causal pathways described previously, a challenge common to the modelling of structural interventions generally [54]. It is also partly due to the lack of consistent evidence of the impact of economic strengthening interventions on clinical outcomes. Agent-based models, which represent individual-level variation in health risk exposure and outcomes, are well-suited to representing complex causal pathways and diverse outcomes, and are increasingly utilised in assessing socio-economic disparities in health [55].

In this study, we make a first attempt at modelling the impact of economic strengthening interventions in South Africa, the country with the world's largest HIV epidemic. Although South Africa has a high gross domestic product per capita, by African standards ($6,253 in 2024 [56]), it also has the highest level of income inequality globally [57], and is therefore an important setting in which to assess the relationship between socio-economic status and reproductive health. Building upon recent advances in the use of agent-based models to represent socio-economic differences in health outcomes [55], we aim to assess which economic strengthening interventions are likely to have the greatest impact on HIV, STIs, and teenage births.

## Methods

### Model structure

We adapted a previously-developed model of HIV and reproductive health in South Africa (MicroCOSM, or Microsimulation for the Control of South African Morbidity and Mortality). This is an agent-based model, which has previously been used to evaluate structural drivers of HIV and STIs [58–60]. Briefly, the model simulates a nationally-representative sample of the South African population, with each individual ('agent') being randomly assigned a set of characteristics: demographic (age, sex, and race), socio-economic (education, urban/rural location, migration, and incarceration history), psychological (gender norms, conscientiousness), healthcare access (use of hormonal contraception, HIV prevention, and HIV testing/treatment), and behavioural (alcohol consumption, propensity for concurrent partners, sexual experience, sexual preference, and current relationship/marital status). Individuals are classified as 'high-risk' or 'low-risk' based on their propensity for concurrent partners, and four types of sexual relationship are modelled (Fig 1A): long-term marital/cohabiting, short-term noncohabiting, casual/once-off (characterised as 'transactional sex') and commercial sex (between

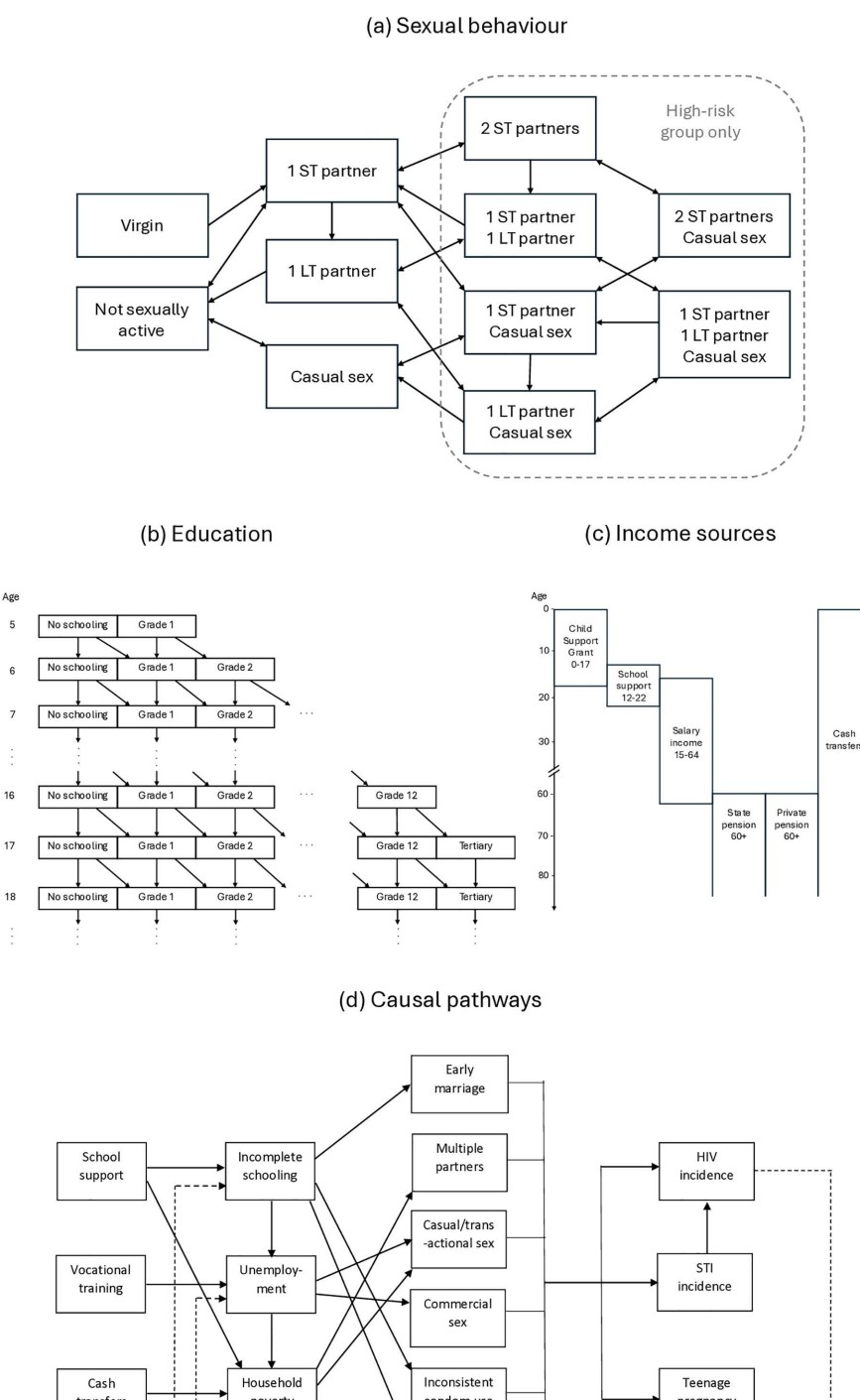

**(a) Sexual behaviour**

**(b) Education**

**(c) Income sources**

**(d) Causal pathways**

**Fig 1. Model structure.** In panel **(a)**, high-risk women can move in and out of sex work and high-risk men can engage in sex with sex workers at any time (not shown); only high-risk individuals can have more than one partner at a time. In panel **(b)**, all individuals currently in school can leave school at any time (not shown), and return to school is permissible only in "School support" intervention scenarios. Panel **(c)** shows how eligibility to receive

different types of income varies in relation to the age of each household member ("School support" and "Cash transfers" only apply in the relevant intervention scenarios). In panel **(d)**, dashed lines represent 'feedback effects'; as described previously [59], girls who fall pregnant while in school are assumed to either drop out of school permanently or to repeat the grade they were in at the time of the pregnancy. LT, long-term (cohabiting/marital), ST, short-term, STI, sexually transmitted infection.

sex workers and clients). When new partnerships are formed, individuals are linked to other individuals in the simulated population, and the transmission of HIV/STIs is modelled based on specified probabilities of transmission per condomless sex act (with multiplier adjustments to account for factors such as the HIV–positive partner's HIV viral load and male circumcision). A woman's monthly probability of conception is similarly based on the woman's numbers of sexual partners, age, and contraceptive use. The simulation begins in 1985, with HIV being introduced into the simulated population in 1990.

Socio-economic status has a number of dimensions, which can influence sexual behaviour in different ways, and socio-economic status is often measured in terms of household wealth/income [61]. We therefore chose to model socio-economic status based on a combination of household- and individual-level characteristics. We extended the model to identify family links between individuals (parents, children, and siblings), and we grouped individuals into households based on family and marital relationships. We modelled household formation and dissolution dynamically, calibrating the model to national survey data on household composition (see section 1.2 of S1 Text).

The modelling of educational attainment has been described previously [59], and is illustrated in Fig 1B. We further extended the model to simulate employment in individuals aged 15–64 who are not currently in school/studying or incarcerated. Rates of entering employment are assumed to depend on age, sex, educational attainment, race, urban/rural location, and parity (in women). In addition, we assumed people living with HIV experience a reduced odds of employment if they are untreated and have low CD4 counts [62–64]. We calibrated the model to national survey data on employment levels by age, sex, and race (see section 1.1 of S1 Text).

We modelled four sources of household income in the baseline scenario: salaries/wages (for each employed member in the household), child support grant payments, the state old age pension, and private pensions (Fig 1C). A more detailed description of these four income sources is provided in section 1.3 of S1 Text. The adjusted per capita household income (APCHI) is calculated as

$$\text{APCHI} = \frac{T}{(n_A + \alpha n_C)^\theta}$$

where $T$ is the total household income, $n_A$ and $n_C$ are the numbers of adults (15+) and children respectively in the household, $\alpha$ is the relative cost of supporting children (relative to adults) and $\theta$ represents the economies of scale in meeting the needs of larger households. In line with previous South African studies [65], we set $\alpha = 0.5$ and $\theta = 0.9$; other equivalence scales are reviewed elsewhere [66].

Fig 1D illustrates the hypothesised causal pathways linking socio-economic status and HIV/reproductive health outcomes. We assumed economic strengthening interventions affect socio-economic variables (educational attainment, employment, and household income), and these in turn affect sexual behaviour; sexual behaviour then influences HIV, STIs, and pregnancy (which can in turn influence socio-economic status, e.g., pregnancy causing school dropout). Table 1 summarises the key parameters that relate socio-economic status to sexual risk behaviour and health seeking behaviour (described more fully in sections 1.4 and 1.5 of S1 Text). We modelled three intervention types, all targeting households with adjusted per capita income below the national average: (1) school support to reduce school dropout (the intervention is also assumed to increase the chance of re-enrolment in youth aged 13–22 who have dropped out of school); (2) vocational training (for ages 20–49); and (3) unconditional cash transfers. We selected these interventions from a more

**Table 1. Model parameters relating socio-economic status to sexual risk behaviour and health seeking behaviour.**

| Model parameters | Mean | SD† | Data sources |
|---|---|---|---|
| Sexual debut | | | |
| Increase in rate of debut in females per log reduction in APCHI* | 0.31 | 0.38 | [31–33,67–70] |
| RR of sexual debut in females if currently in school | 0.46 | 0.44 | [69,71–77] |
| RR of sexual debut in males if currently in school | 0.80 | 0.45 | [71,73] |
| School dropout | | | |
| Probability of permanent dropout in the year of giving birth | 0.35 | – | [78] |
| Casual/transactional sex | | | |
| Increase in casual sex in females, per log reduction in APCHI* | 0.43 | 0.69 | [38–40,79] |
| Increase in casual sex in men who are employed (vs unemployed) | 0.32 | 0.54 | [32,35,79] |
| Commercial sex | | | |
| RR of commercial sex in men who are employed | 1.40 | 0.63 | [36,80,81] |
| RR of commercial sex in women who are employed | 0.00 | – | [82] |
| Short-term, noncohabiting relationships | | | |
| RR of partner acquisition in men who are employed | 1.32 | 0.43 | [31,37,79] |
| Marital and cohabiting relationships | | | |
| RR of marriage if currently in school | 0.37 | 0.34 | 1993 OHS |
| RR of male marriage if completed high school (no tertiary) | 1.50 | – | [83] |
| RR of male marriage if completed tertiary education | 3.50 | – | [83] |
| RR of female marriage if completed high school | 0.70 | – | [27,83,84] |
| Condom use | | | |
| OR for consistent condom use per year of schooling | 1.05 | 0.08 | [85,86] |
| Inequitable gender norms | | | |
| RR of endorsing inequitable gender norms if tertiary educated | 0.51 | – | [58] |
| RR of casual sex in men per 0.1 reduction in inequitable gender norms | 0.81 | – | [58] |
| RR of concurrent partners in men endorsing inequitable gender norms | 3.18 | – | [58] |
| RR of condom use in men endorsing inequitable gender norms | 0.56 | – | [58] |
| Health seeking behaviour | | | |
| RR of medical male circumcision per log increase in APCHI* | 1.15 | 0.19 | [41,42] |
| RR of HIV testing per year of education | 1.12 | – | [87] |
| RR of hormonal contraceptive use per year of education | 1.15 | – | [85] |
| Intervention effectiveness | | | |
| RR of school dropout if receiving school support (nonmaterial) | 0.68 | 0.20 | [88] |
| RR of school dropout per R800 of school support | 0.86 | 0.10 | [89] |
| Probability of return to school if aged 13–22 and receiving school support | 0.34 | – | [69] |
| OR of unemployment if receiving vocational training/microfinance | 0.87 | 0.10 | [90] |

APCHI = adjusted per capita household income; OHS = October Household Survey; OR = odds ratio; RR = relative rate; SD = standard deviation.

*Per unit difference between the natural log of the APCHI and the log of the national average APCHI, for households that have a log APCHI below the national average (for those above the average, no income effect is modelled).

† Standard deviations are specified for the parameters that are varied in the model calibration process (sections 1.4–1.6 of S1 Text describe the prior distributions in more detail); for all other parameters, the value is fixed.

complete list of economic strengthening interventions [45], based on the availability of data regarding their impacts on HIV/STI/teenage pregnancy outcomes. The school support intervention is assumed to include both material support (e.g., school uniforms, cost of school transport) and nonmaterial support (e.g., attendance monitoring and counselling), both of

which reduce the probability of school dropout. We modelled the effects of the two components separately, with half of the cash equivalent value of the material support being added to the household income, in the same way as for cash transfers. We assumed vocational training increases the odds of finding employment in currently unemployed individuals. Cash transfer amounts are added to total household income as another source of income.

## Model calibration

We adopted a Bayesian approach to model calibration [91], similar to that in a previous application of our model [58]. Prior distributions were specified to represent the uncertainty in key model parameters (Table 1). We specified hurdle distributions to represent the uncertainty in the effect of socio-economic status on sexual risk behaviour, with the hurdle representing a nonzero probability of a null relationship (this was included to avoid excessive prior confidence in the purported causal relationship). We set the nonzero probabilities based on a review of the strength of evidence from RCTs; a more detailed description of the prior distributions is provided in section 2.1 of S1 Text.

We calibrated the model to the results of RCTs of economic strengthening interventions conducted in sub-Saharan Africa. RCTs were identified from recent reviews of economic strengthening interventions [45,46,48,49]. We classified included trials as being either 'pure' cash transfer interventions (including both conditional and unconditional cash transfers, but without any strong conditioning on school attendance), school support interventions (which typically aimed to promote school retention, often through the provision of financial support, or through cash transfers conditional on school attendance) and vocational training programmes (directed to individuals who were out of school, providing training to improve their employment prospects and/or credit to enable them to establish their own business). We excluded less frequently studied interventions (e.g., savings programmes, food assistance, and financial education) due to limited RCT data available on their effects [45]. Intervention effects were calculated on a natural log scale, with associated standard errors (Table ZB in S1 Text).

A sample of 5,000 parameter combinations was randomly drawn from the prior distributions in Table 1. For each parameter combination, we ran the model eight times: twice in the 'baseline' scenario, twice in the cash transfer scenario, twice in the school support scenario, and twice in the vocational training scenario (two simulations being necessary in order to quantify the extent of stochastic variation in model outputs; the average of the two results was calculated for the purpose of estimating intervention effects). For the purpose of calibration to historic RCT data, we assumed all interventions started in 2005 (close to the average start year of the included RCTs), and we set the annual value of the cash transfer in 2005 to R800 (equivalent to $117 in 2005), the average value of the cash transfers in the included RCTs. We assumed the interventions were directed to households with log APCHI below the national average (8.68 in 2005, equivalent to R5856 or $861 annually). The modelled intervention effect was calculated as the log of the odds ratio/relative risk for the outcome of interest (comparing the intervention and baseline scenarios). We calculated a likelihood value by comparing the modelled intervention effect and the reported intervention effect, assuming the log difference followed a normal distribution with zero mean. Log likelihood values were summed for all outcomes, across all RCTs, to calculate a total likelihood for each parameter combination; separate likelihoods were also calculated for each of the three intervention types.

We drew a posterior sample of 100 parameter combinations from the initial set of 5,000 parameter combinations, using the likelihood values as weights. This sample was used to calculate the posterior means and 95% confidence intervals (CIs), running the model five times for each parameter combination to reduce stochastic variation.

We validated the model by comparing the modelled odds ratios for the associations between socio-economic status and HIV/sexual risk behaviour, as observed in four nationally-representative household surveys in 2005, 2012, 2016, and 2017 [79,92].

In all scenarios, we set STI transmission probabilities at the median of the distribution of best-fitting parameters identified in previous model calibrations [93]. We adjusted HIV transmission probabilities per sex act to yield plausible estimates of HIV incidence and prevalence (see section 3.2 of S1 Text). We estimated teenage fertility rates, per year

of sex with a single partner in the absence of contraception, from age- and race-specific fertility rates in the baseline scenario [94].

**Population attributable fractions**

We calculated the proportion of incident HIV cases/STIs/teenage births that were attributable to low socio-economic status, over the 2000–2020 period, by running a counterfactual scenario in which the sexual behaviour and health seeking behaviour of all individuals are the same as might be expected (a) if they had completed tertiary education, (b) if they remained in education at least to age 21, (c) if they were employed, and (d) if their adjusted per capita household income were no lower than the national mean in the baseline scenario (with the changes in behaviour occurring from mid-2000). We calculated the population attributable fraction (PAF) as the proportionate difference in cumulative HIV cases/STIs/teenage births, over the 2000–2020 period, between this counterfactual scenario and the baseline scenario. Partial rank correlation coefficients (PRCCs) were calculated to assess associations between PAFs and each of the parameters that were changed in the counterfactual scenario [95].

**Future intervention scenarios**

We modelled three future intervention scenarios: school support, vocational training, and cash transfers. Future intervention impacts were considered over the period from 2025 to 2040. We assumed interventions were limited to households with adjusted per capita household income below the national average in the baseline scenario. We set the annual value of the cash transfer to $227 per household in the case of cash transfer interventions and $114 per eligible youth in the case of school support (at 2023 exchange rates), increasing in line with inflation [96].

All model code and data inputs are available from https://github.com/leighjohnson/MicroCOSM.

# Results

We present the results in two parts, the first describing the results of the model calibration and the role of socio-economic status in the period up to 2020, and the second describing possible future changes if economic strengthening interventions were to be introduced.

## Calibration to economic and reproductive health data

Our model was in reasonable agreement with observed HIV prevalence and incidence trends in South Africa, external estimates of the incidence of gonorrhoea and chlamydia [97] and data on teenage fertility (Fig 2A–2D). The model was also in agreement with census and survey data showing rising levels of high school completion among youth, and stable employment levels (Fig 2E and 2F). The model also estimated rising income together with declining income inequality, the result of increasing levels of social welfare expenditure, and both trends were validated by estimates from Statistics South Africa (Fig 2G and 2H).

## Calibration to intervention effectiveness data

Posterior estimates of model parameters were largely consistent with prior distributions, although intervention effect parameters differed significantly: vocational training was estimated to be less effective in reducing unemployment (posterior odds ratio [OR] 0.93, prior OR 0.87), while posterior estimates of school support impacts were mixed, with material support being more effective, compared to prior assumptions about effects on school dropout (Table ZD in S1 Text). The posterior estimates of RCT effects were in good agreement with the data, although the model was unable to match the large reductions in HIV and herpes prevalence and pregnancy incidence in one trial [69] (Figs N–P in S1 Text). The model was also consistent with most of the validation data, although the model under-estimated the strength of the negative association between HIV prevalence and education in women (Fig Q in S1 Text).

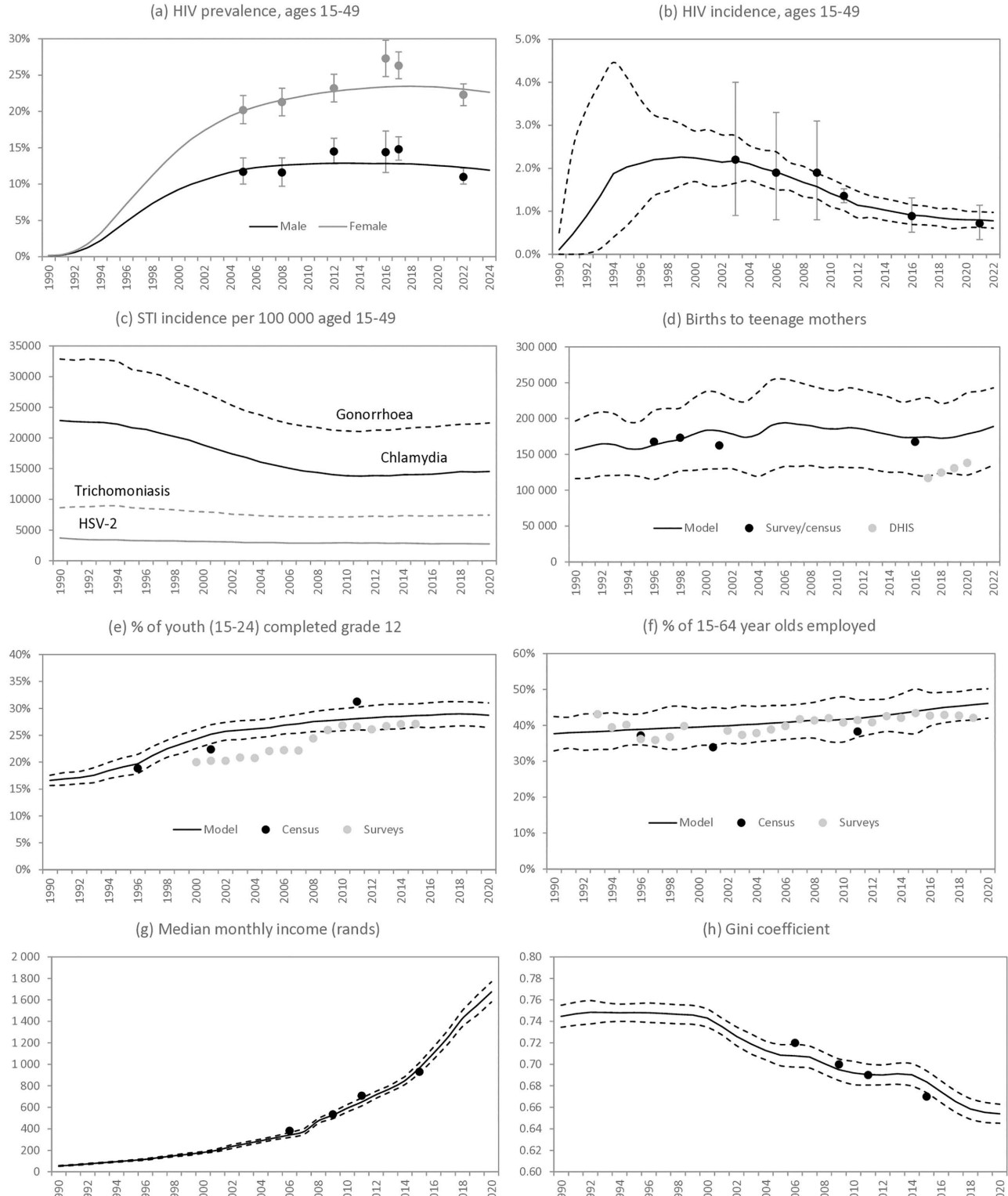

**Fig 2. Model calibration.** Model estimates of key reproductive health outcomes and economic indicators are compared against calibration targets. Reproductive health outcomes include HIV prevalence (panel **a**) and HIV incidence (panel **b**), both in 15−49 year olds, STI incidence per 100,000 population aged 15−49 (panel **c**), and total births to teenage mothers (panel **d**). Economic indicators include the proportion of youth who have completed high

school (panel **e**), the proportion of 15−64 year olds who are employed (panel **f**), median monthly household income per household member (panel **g**) and Gini coefficient (panel **h**). Except in panel c, solid lines represent posterior mean model estimates and dashed lines represent 95% confidence intervals. Calibration data points in panels a and b are from national household surveys [79,92,98], while data in panel d are from the DHIS [99], censuses and surveys [79,100]. Data in panels e and f are from censuses, October Household Surveys, General Household Surveys and Labour Force Surveys. Data in panels g and h are derived from national surveys [101]. DHIS, District Health Information System; HSV-2 = herpes simplex virus type 2; STI, sexually transmitted infection.

**Table 2. Proportion of incident HIV cases, STIs, and teenage births attributable to low socio-economic status, 2000–2020.**

|  | HIV | STIs | Teenage births |
|---|---|---|---|
| Total | 13% (1,50%) | 7% (−1,19%) | 31% (10,76%) |
| Males | 11% (−1,49%) | 6% (−2,19%) | − |
| Females | 14% (0,50%) | 8% (−1,20%) | 31% (10,76%) |

**Table 3. Correlates of population attributable fractions.**

| Parameter | HIV | STIs | Teenage births |
|---|---|---|---|
| OR for consistent condom use per year of schooling | **0.87** | **0.90** | **0.57** |
| Increase in casual sex in men who are employed | −0.09 | −0.05 | −0.01 |
| RR of commercial sex in men who are employed | 0.04 | −0.16 | −0.11 |
| Increase in casual sex in females, per log reduction in APCHI* | −0.14 | −0.04 | −0.12 |
| RR of partner acquisition in men who are employed | **0.30** | **0.27** | **0.27** |
| Increase in sexual debut in females per log reduction in APCHI* | 0.13 | 0.15 | 0.07 |
| RR of medical male circumcision per log increase in APCHI | −0.08 | −0.09 | 0.01 |
| RR of sexual debut in females if currently in school | **−0.63** | **−0.87** | **−0.95** |
| RR of sexual debut in males if currently in school | −0.05 | −0.01 | **−0.27** |

Values are partial rank correlation coefficients between the input parameters (Table 1) and the population attributable fractions (Table 2); values in bold indicate correlation coefficients that are statistically significant at the 5% level.

APCHI = adjusted per capita household income; OR = odds ratio; RR = relative rate.

*Per unit difference between the natural log of the APCHI and the national average log APCHI, for households that have a log APCHI below the national average (for those above the average, no income effect is modelled).

## Population attributable fractions

Low socio-economic status accounted for 13% of new HIV infections, 7% of incident STIs (gonorrhoea, chlamydia, and trichomoniasis) and 31% of teenage births in South Africa, over 2000−2020 (Table 2). However, because of uncertainties regarding effect sizes, CIs around these PAFs were wide (1,50% for HIV; −1,19% for STIs; and 10,76% for teenage births). PAF estimates were similar for men and women. Table 3 shows the socio-economic effect parameters that account for the most uncertainty. The effect of education on condom use was the most significant correlate of the HIV PAF (PRCC = 0.87) and the STI PAF (PRCC = 0.90), but was less significant as a correlate of the teenage birth PAF (PRCC = 0.57). The relative rate of female sexual debut while in school was the most significant correlate of the teenage birth PAF (PRCC = −0.95) and was also significantly correlated with the HIV and STI PAFs; the relative rate of male sexual debut while in school was also weakly associated with the teenage birth PAF. The effect of male employment on partnership formation was positively associated with all the PAFs, although this was only of borderline significance. Other parameters were not statistically significant, and fewer parameters were significant when standard correlation coefficients were calculated in place of PRCCs (Table ZG in S1 Text).

## Intervention impacts

In the absence of any economic strengthening interventions, our model suggests there would be little change in income inequality and high school completion over the next 15 years, although employment is expected to increase modestly as a result of population ageing (Fig 3). The cash transfer intervention would reduce income inequality, with the Palma ratio (the ratio of income in the top decile to that in the four lowest deciles) decreasing from 8.5 to 7.8 by 2030; school support and vocational training would have more modest effects on income inequality (Fig 3A). The school support intervention would significantly increase the proportion of South African youth who have completed high school, from 30% to 38% by 2040, but the other interventions would have no effects on school completion (Fig 3B). Vocational training interventions would only slightly increase employment levels, while school support would

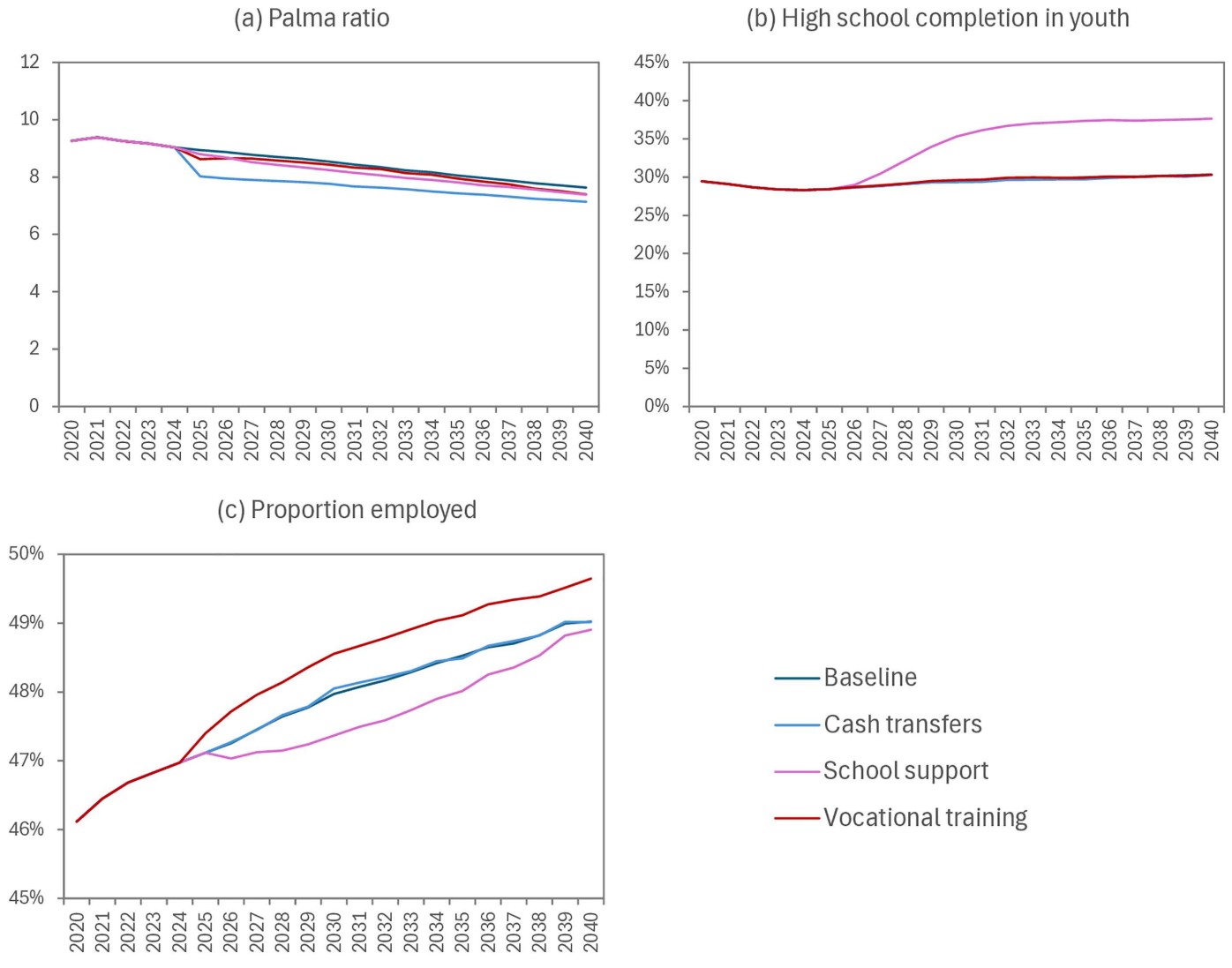

**Fig 3. Intervention impacts on socio-economic outcomes.** The Palma Ratio (panel A) is defined as the ratio of total income in the top income decile to the total income in the four lowest deciles. In panel (b), the high school completion fraction is the proportion of all 15–24 year old South Africans who have successfully completed grade 12. In panel (c) the denominator is all individuals aged 15–64 (including those in school and those not actively seeking work).

slightly reduce the fraction of the population employed in the short-term, due to young adults remaining in school for longer (Fig 3C).

When considering intermediate behavioural outcomes, the effects of the cash transfer and vocational training interventions are close to zero (Table 4). However, the school support intervention is expected to significantly reduce the proportion of sexually active young females who are not using hormonal contraception, the proportion of adults in concurrent partnerships, and the proportion of men engaging in casual sex (the latter two effects being due to assumed effects of education on inequitable gender norms, and assumed effects of inequitable gender norms on male risk behaviour [58]). The school support intervention may also reduce the proportion of adolescents who are sexually experienced and the number of early marriages, although these reductions are not statistically significant (Table 4).

Our model suggests that economic strengthening interventions would have no significant impact on most reproductive health outcomes (Table 4). Over 2025–2040, schooling support is the intervention that would have the greatest impact, with an anticipated 5% reduction in teenage births (which is strongly correlated with the relative rate of sexual debut for girls who remain in school [PRCC = −0.84]) and a 1% reduction in HIV and STI incidence (which is strongly correlated with the relative rate of sexual debut among girls in school [PRCC = −0.58 for STIs and −0.27 for HIV] and the increase in condom use per year of education [PRCC = 0.58 for STIs and 0.25 for HIV]). For all other intervention/outcome combinations, the expected impact is 0%, with lower CI limits below zero (partly a reflection of residual stochastic variation). Vocational training is unlikely to have a significant impact if it is limited to females (for example, a 0% reduction [95% CI: −1,1] in STIs). Intervention impacts are similar in males and females. Proportionate reductions in HIV, STIs and teenage pregnancies are also similar over the 2025–2040 period, i.e., there is no suggestion of the intervention impact waning or improving at longer durations (Table ZH in S1 Text).

**Table 4. Reduction in behavioural risks and reproductive health outcomes (2025–2040) as a result of different economic strengthening interventions.**

|  | School support | Vocational training | Cash transfers |
|---|---|---|---|
| Behavioural outcomes |  |  |  |
| Sexually experienced adolescents | 4% (0, 10%) | 0% (−1,1%) | 0% (−1,1%) |
| Sexually active females (15–24) notusing hormonal contraception | **1% (0,2%)** | 0% (−1,1%) | 0% (−1,1%) |
| Married youth (15–24) | 3% (−3,9%) | 0% (−6,4%) | 0% (−6,4%) |
| Condomless sex (ages 15–49) | 0% (0,0%) | 0% (0,0%) | 0% (0,0%) |
| People in concurrent partnerships | **1% (0,2%)** | 0% (−1,1%) | 0% (−1,1%) |
| Client contacts with sex workers | 0% (−2,1%) | 0% (−1,1%) | 0% (−1,1%) |
| Men having casual sex | **2% (1,4%)** | 0% (−1,1%) | 0% (−1,1%) |
| Women having casual sex | 0% (−1,2%) | 0% (−1,1%) | 0% (−1,1%) |
| Reproductive health outcomes |  |  |  |
| New HIV infections (ages 15+) | 1% (−3,4%) | 0% (−3,3%) | 0% (−4,4%) |
| New HIV infections (men) | 1% (−3,5%) | 0% (−4,5%) | 0% (−4,4%) |
| New HIV infections (women) | 1% (−4,5%) | 0% (−3,4%) | 0% (−4,3%) |
| New STIs (ages 15+) | 1% (0,2%) | 0% (−1,1%) | 0% (−1,1%) |
| New STIs (men) | 1% (0,2%) | 0% (−1,1%) | 0% (−1,1%) |
| New STIs (women) | 1% (0,2%) | 0% (−1,1%) | 0% (−1,1%) |
| Births to teenagers | 5% (−1,12%) | −1% (−15,3%) | 0% (−3,3%) |

95% confidence intervals are shown in brackets, and significant reductions are formatted in bold.

## Discussion

Although our model suggests a potentially significant contribution of low socio-economic status to the incidence of HIV, STIs, and teenage pregnancy in South Africa, there is substantial uncertainty around the magnitude of this contribution. This uncertainty arises because socio-economic status affects sexual behaviour in diverse ways. On the one hand, our model identifies significant positive effects of the relationship between condom use and educational attainment and the relationship between schooling and sexual debut. On the other hand, increases in male socio-economic status could increase HIV and STI incidence because of the effect of male employment on entry into casual and commercial sex relationships. Because of these offsetting effects, the net effects of economic strengthening on sexual and reproductive health outcomes can be modest.

One possible response to this complexity would be to target economic strengthening to those groups that are most likely to be positively affected, for example, economically vulnerable adolescent girls and young women. While this may have merit, our simulations suggest that vocational training targeted to women (for example) would not significantly alter HIV and STI incidence. We also find that economic strengthening interventions have similar effects in men and women, despite very different socio-economic effects on sexual behaviour. This is largely because HIV and STI transmission in South Africa is predominantly heterosexual (i.e., any gain by one sex in the short-term is likely to result in reduced secondary transmission to the other sex in the longer-term).

The disappointing lack of impact of economic strengthening interventions on reproductive outcomes is to some extent a reflection of the difficulty in achieving significant changes in socio-economic status, against a background of extreme economic inequality. For example, vocational training interventions would increase employment levels by only 1%, and cash transfers to the poorest South African household would only reduce the Palma ratio of inequality by 8% (Fig 3). These modest effects on socio-economic indicators are further diluted at each subsequent step in the causal pathway, so that the final effects on reproductive health outcomes are negligible. More radical structural changes may be needed to achieve substantial gains in economic and reproductive health outcomes. When modelling the future rollout of economic strengthening interventions, we have optimistically assumed that all eligible individuals would receive the intervention and achieve levels of exposure similar to those in RCTs. This assumption is unrealistic, but the nonsignificance of the simulated intervention impacts, despite the optimistic assumptions, reinforces the point that they are unlikely to have substantial impacts in the South African setting.

A number of other factors may explain the nonsignificant intervention impacts. Firstly, we used hurdle distributions to represent prior uncertainty in key parameters, which is conservative because a substantial probability is assigned to a null relationship between each socio-economic variable and sexual risk behaviour. Secondly, and closely related, the RCT data to which we are calibrating the model are too heterogeneous and too imprecise (in most instances) to pull the posterior means away from the prior means (hence there are relatively few cases of significant differences between the prior and posterior estimates in Table ZD in S1 Text). Thirdly, the model is agent-based and therefore stochastic, and stochastic variation also contributes to the imprecision.

There have been few previous attempts to develop mathematical models of the structural drivers of sexual risk behaviour [53], and even fewer that have specifically assessed the role of socio-economic status. A few studies have modelled the effects of poverty and social protection in Brazil [102,103], but these are calibrated based on observed temporal associations between economic indicators and HIV indicators, not on the results of individual-level associations or RCTs of economic strengthening. A strength of our calibration approach is that we have used a Bayesian approach to combine both local observational data on the likely relationships between socio-economic status and sexual behaviour (through prior distributions) and RCT data on the impacts of economic strengthening interventions in African settings (through a likelihood function). It is perhaps disappointing that the resulting CIs around the posterior model estimates remain wide, despite the systematic approach to including different types of evidence, but we believe this reflects real uncertainty around socio-economic relationships. Due to the substantial computing time required for each simulation, it was not

feasible to draw a larger posterior sample, which would be important in obtaining more accurate 95% CIs. The 95% CIs should therefore be interpreted with caution.

A limitation of this model is that we have focussed mainly on the effects of socio-economic status on sexual behaviour and have ignored some of the other effects that are relevant to sexual and reproductive health outcomes. For example, we have not modelled effects of socio-economic status on mortality in people living with HIV [43]. We have also not modelled an effect of socio-economic status on STI health seeking [104], although in the South African setting higher socio-economic status is not necessarily associated with better STI treatment [105,106]. We have nevertheless included socio-economic effects on hormonal contraceptive use, male circumcision and rates of HIV testing, all of which are important. We have not modelled some of the more detailed causal pathways that link socio-economic status and sexual risk behaviour. For example, there is evidence of a relationship between food insecurity and sexual risk behaviour, particularly in women [39,107–109]. In addition, poverty is associated with greater mental distress [110–112], which in turn may be linked to higher sexual risk behaviours [111,113,114]. Education is also associated with exposure to life skills programmes and HIV prevention messaging [115].

Another limitation is that we have not modelled the effects of COVID-related lockdowns on education [116], employment [117], antiretroviral treatment (ART) uptake [118], and contraceptive use [119]. The adverse social effects of the COVID pandemic were to some extent mitigated by the introduction of a Social Relief of Distress grant, which could be accessed by adults who were unemployed and not receiving any other income [120], and this grant has been continued into the post-COVID period. There have been calls to make this grant permanent, and to increase the amount of the grant [120]. A detailed analysis of the impact of COVID on sexual and reproductive health is beyond the scope of this study, as is the analysis of the potential impact of the Social Relief of Distress grant and its continuation. However, it is worth noting that in our cash transfer scenario we have considered a grant that is worth $227 per annum in 2023, which is similar to the $223 per annum currently under the Social Relief of Distress grant. Given the modest HIV and STI impacts estimated in our cash transfer scenario, it seems unlikely that the Social Relief of Distress grant has had much impact on sexual and reproductive health outcomes.

Other limitations should be considered. Our analysis relies only on publicly available data, but additional longitudinal individual-level data, such as collected in Health and Demographic Surveillance Systems [121], may be valuable in validating the model assumptions about effects of socio-economic status on sexual behaviour, fertility, and HIV. The model also does not represent the effect of social networks on sexual behaviour [122], and the formation of these networks is likely to be heavily influenced by education and employment. The model does not represent intergenerational transfers in education [123], and a 15-year projection term is likely to be insufficient for representing the longer-term benefits of investment in education. Different measures of socio-economic status can influence risk behaviour in different ways, but in the interests of model parsimony we have limited our model to one socio-economic effect on each risk behaviour. There is also uncertainty regarding the relative importance of absolute and relative measures of deprivation in driving risk behaviour [124,125]; in our model we assume a combination of both. In line with 'Diffusion of innovations' theory [126] and the 'Inverse equity hypothesis' [127], we have assumed increases in condom use in the early stages of the HIV epidemic follow a strong socio-economic gradient, but that this gradient reduces over time; further work is required to assess the validity of this assumption, and whether it applies to other HIV interventions. These complexities are discussed in further detail in section 1.4 of S1 Text.

This study considers the role of socio-economic factors at the individual and household level, but socio-economic characteristics of the community may also be important in driving sexual risk behaviour, even when controlling for individual socio-economic status [128,129]. In South Africa, income inequality at the municipal level significantly increases women's risk of HIV [130], and income inequality has also been shown to explain much of the variation across countries in sexual risk behaviour [131], HIV prevalence [21,132,133], and teenage fertility [3]. Further work is required to understand (and potentially model) these community effects. Our results might not be generalisable to other African settings,

PLOS Medicine

where there is typically less income inequality but more absolute poverty. In a recent analysis of national household survey data, HIV prevalence was inversely associated with education in southern and East Africa, but not in West and Central Africa [16]. Unlike in southern Africa, ART coverage and socio-economic status were positively associated in other African regions [16].

In the short-term, and in the context of a dedicated but declining budget for HIV programmes, it would make sense to focus HIV spending on interventions that have been shown to be highly cost-effective in reaching HIV endpoints (e.g., condom distribution, ART, and pre-exposure prophylaxis for key populations [134]), rather than economic strengthening. However, economic interventions can have important positive impacts beyond sexual and reproductive health, including on mental health [76,135–137], tuberculosis [138], child health [139], and social well-being [140]. Ideally, all of these outcomes should be considered when evaluating the impact or cost-effectiveness of economic interventions, and we do not suggest that policy decisions should be based on sexual and reproductive health outcomes alone. There is a need for a more inter-sectoral 'whole of government' approach to addressing health challenges [141], and this is especially true when considering economic strengthening interventions. This necessitates more advanced analytic tools, both in cost-effectiveness evaluation [142,143] and in simulation of complex systems [55,144]. This study represents a first step towards capturing this inter-sectoral complexity in a model of a middle-income country with significant income inequality.

## Supporting information

**S1 Text. Supplementary material.**
(PDF)

## Acknowledgments

The authors thank Marie Stoner for helpful comments on an earlier draft.

The conclusions and opinions expressed in this work are those of the authors alone and shall not be attributed to the Gates Foundation.

## Author contributions

**Conceptualization:** Leigh F. Johnson, Lise Jamieson, Gesine Meyer-Rath.

**Formal analysis:** Leigh F. Johnson.

**Funding acquisition:** Leigh F. Johnson.

**Investigation:** Lise Jamieson, Mmamapudi Kubjane.

**Methodology:** Leigh F. Johnson, Gesine Meyer-Rath.

**Writing – original draft:** Leigh F. Johnson.

**Writing – review & editing:** Leigh F. Johnson, Lise Jamieson, Mmamapudi Kubjane, Gesine Meyer-Rath.

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
