## [Editor Report · Decision Letter 0]

29 May 2025

Dear Dr Johnson,

Thank you for submitting your manuscript entitled "Money, jobs or schooling? A model-based evaluation of economic strengthening in South Africa and its impact on HIV, sexually transmitted infections and teenage births" for consideration by PLOS Medicine.

Your manuscript has now been evaluated by the PLOS Medicine editorial staff as well as by an academic editor with relevant expertise and I am writing to let you know that we would like to send your submission out for external peer review.

Please include the appropriate checklist for this modeling study (see https://www.equator-network.org/reporting-guidelines/), and ensure that you list the RCTs used in the model calibration in a distinct supplementary table.

For clinical studies, please upload a copy of your trial study protocol as a supporting information file. The study protocol should be the version submitted for approval to the institutional review board or ethics committee, should include any amendments to the study protocol, as well as the date of their approval by the institutional review or ethics committee. Please also detail any deviations from the study protocol in the Methods section of your manuscript. The editors will consider the protocol and study conduct prior to a final decision for external review.

Please re-submit your manuscript within two working days, i.e. by Jun 02 2025 11:59PM.

Kind regards,

Alison Farrell, Ph.D.

Senior Editor

PLOS Medicine

---

## [Decision Letter · Decision Letter 1]

29 Jul 2025

Dear Dr Johnson,

Many thanks for submitting your manuscript "Money, jobs or schooling? A model-based evaluation of economic strengthening in South Africa and its impact on HIV, sexually transmitted infections and teenage births" (PMEDICINE-D-25-01888R1) to PLOS Medicine. The paper has been reviewed by subject experts and a statistician; their comments are included below and can also be accessed here: [LINK]

As you will see, the reviewers the topic important and the findings of interest. After discussing the paper with the editorial team and an academic editor with relevant expertise, I'm pleased to invite you to revise the paper in response to the reviewers' comments. We plan to send the revised paper to some or all of the original reviewers, and we cannot provide any guarantees at this stage regarding publication.

As you will see in their remarks, the reviewers find that the study requires substantial revision to provide more details on the modeling, underlying rationale for choices and additional information to allow replication of the results. A more extensive discussion of the study limitations and the implications of the findings is also needed, along with additional analyses and ideally external validation. Further note that the code, code, calibration data, and simulation instructions are required in the revised manuscript and we ask that you deposit the code in a public repository. Please also see the comments below of the academic editor.

We ask that you submit your revision by Aug 19 2025 11:59PM. However, if this deadline is not feasible, please contact me by email, and we can discuss a suitable alternative.

Don't hesitate to contact me directly with any questions (afarrell@plos.org).

Best regards,

Alison

Alison Farrell, Ph.D.

Senior Editor

PLOS Medicine

afarrell@plos.org

Comments from the academic editor:

The methods could be a bit better organized to reflect the main output/results. It does seem that two of the main purposes was to evaluate population attributable fractions for various socioeconomic factors and then also the effects of economic empowerment interventions.

Please include the proximal, intermediate, and distal impacts of the economic empowerment interventions. Currently the intermediate is missing (e.g., types of partners, condom use, type of sex, etc.).

In Figure 3, although the most likely affected metric is shown for each type of intervention, it would be worthwhile to have all 3 on each of the panels. For example, schooling and vocational training could also be expected to affect income.

Comments from the reviewers:

Reviewer #1: Major Comments:

a) The MicroCOSM model is described in general terms in both the manuscript and supplementary materials, but key submodules (e.g., for education, income, sexual behaviour) lack sufficient detail for replication. I recommend including a summary table listing key inputs, outputs, and governing rules for each sub-model, possibly with illustrative pseudo-code or flow diagrams.

b) The model's assumptions regarding behavioural mechanisms (e.g., how socioeconomic status impacts condom use, sexual debut, concurrency) should be described with greater precision, including any evidence base supporting these pathways.

c) The combination of trial-derived and observational data for prior distribution specification is methodologically reasonable. However, several priors appear overly narrow or based on limited data, without adequate justification for the selected bounds. A clearer rationale for the shape and truncation of priors, supported by references, should be included in the supplementary materials.

d) The use of hurdle priors (Section 2.1 in supplementary file) to reflect uncertainty around null effects is innovative, but their functional form and probability mass allocation are not fully explained. A brief sensitivity analysis on the choice of hurdle probabilities would be valuable.

e) The Bayesian calibration approach is appropriate, but the likelihood construction is insufficiently described. Details should be added on how different trial outcomes were weighted in the composite likelihood, variance assumptions for the outcome measures, and any convergence diagnostics or posterior predictive checks performed.

f) Only 100 simulations were drawn from the posterior sample. This seems modest given the dimensionality of the model. The authors should comment on the adequacy of this sample size for uncertainty quantification and consider increasing it.

g) The assumptions underlying the implementation of interventions (e.g., universal access, full adherence, constant effect sizes) may not reflect real-world feasibility. I recommend including more realistic scale-up scenarios or at least discussing this as a limitation.

h) The counterfactual scenario—where all individuals attain tertiary education, secure employment, and earn above the national income average—appears implausibly optimistic. This may bias the estimated population attributable fractions (PAFs) upward. A more conservative or gradient-based counterfactual would improve credibility.

i) The model treats changes in socioeconomic status.as exogenous to behavioural risk, but real-world interactions (e.g., pregnancy affecting education, HIV status affecting employment) suggest bidirectional causality. These feedbacks are not captured in the current model and should be acknowledged explicitly.

j) Trial-derived effect estimates are applied nationally, but trials such as HPTN 068 and Swa Koteka may not reflect national demographic or geographic variation. The authors should address how they accounted for the limited generalisability of trial populations in model calibration.

k) While the model is validated against national survey data, reliance on cross-sectional associations is limiting. Additional external validation (e.g., longitudinal data, cohorts not used in calibration) would improve credibility.

l) The authors may also wish to discuss whether their findings are transferable to other sub-Saharan African contexts, and under what conditions such generalisation might be appropriate.

m) The model code is stated to be available upon acceptance. In line with open science standards, I strongly recommend that the code, calibration data, and simulation instructions be made available at the time of peer review to enable full reproducibility.

Minor Comments:

n) The adjusted per capita household income (APCHI) scale is used without discussion of alternatives (e.g., OECD-modified scale).

o) Income is modelled as relatively stable, which may understate income volatility and its effects on behaviour. If dynamic income variation is not modelled, this should be acknowledged.

p) It is unclear how the model handles uncertainty in parameters derived from heterogeneous or sparse data sources. A table mapping source type to uncertainty specification would improve transparency.

q) Behavioural outcomes (e.g., concurrency, condom use) may be sensitive to social network structures and local norms, which are not accounted for in the current individual-level modelling. This could be discussed as a potential limitation.

r) Time horizon of 15 years may be insufficient to capture the full long-term or intergenerational benefits of economic interventions, especially those affecting educational attainment. This limitation should be acknowledged.

Reviewer #2: This manuscript presents a valuable effort to model the complex relationship between economic strengthening interventions and HIV, STIs, and teenage pregnancy in South Africa. The use of an agent-based model (MicroCOSM) is commendable, and the study attempts to quantify the potential impact of various interventions. However, several aspects require further clarification and justification. The limited impact of the modeled interventions is a significant finding, but the reasons behind this need to be explored more thoroughly.

1. "Over 2025-2040, none of the interventions are estimated to have impacts significantly different from zero, due to limited impact on secondary economic outcomes." This is not clear and misleading. What are the interventions? The methods part should highlight this.

2. The wide confidence intervals for the PAF estimates should be highlighted as a key limitation.

3. The rationale for using a modeling approach should be clearly stated. What specific questions can the model address that cannot be answered through empirical studies alone?

4. The introduction should also emphasize what gaps this modeling study aims to fill and how it builds upon existing literature.

5. The complexity of education and social economics on HIV and STIs may be relevant to different factors, i.e., rural or urban, and gender. The modeling should also take these into account, especially given no clear conclusions between those with HIV and STIs.

6. What are the key assumptions of the model? How were these assumptions validated? The results section should report the validation results.

7. A flow diagram illustrating the key components of the model and their interactions would be helpful.

8. The reliance on publicly available data is understandable, but the limitations of this data should be acknowledged.

9. The process of calibrating the model to RCT data needs to be described in more detail. How were the various RCTs selected? How were the intervention effects translated into model parameters? What measures were taken to address heterogeneity across the RCTs?

10. The rationale for using hurdle distributions for the socio-economic status parameters should be explained more fully.

11. The choice of the three intervention types (school support, vocational training, and unconditional cash transfers) needs to be justified. Are these the most promising or widely implemented economic strengthening interventions in South Africa?

12. The assumptions about how the interventions affect model parameters should be clearly stated.

13. The methods for calculating PAFs are described.

14. The finding that none of the interventions have a significant impact over 2025-2040 is surprising. The reasons for this lack of impact should be explored in more detail. Is it due to the limited reach of the interventions, the complexity of the causal pathways, or limitations in the model? In the methods section, the author should mention the impact of the selected interventions from previous results/models.

15. The discussion should also consider the policy implications of the findings. If economic strengthening interventions are unlikely to have a significant impact on HIV, STIs, and teenage pregnancy, what other strategies should be pursued?

16. The authors need to acknowledge that social and environmental factors will also contribute to social wellbeing.

Reviewer #3: Leigh Johnson and colleagues used a previously developed agent based model to investigate the role of socioeconomic factors that drive South Africa's high incidence rates of HIV, STI and teenage pregnancies. Examining these socioeconomic factors within a model is important due to the complex relationships between these socioeconomic factors. The agent-based model that was used included the effects of education, employment and per-capita household income on sexual behaviours. Using the model the authors found that that low socioeconomic status accounted for 14% of new HIV infections, 10% of STIs, and 46% of teenage births from 2000 to 2020. However, because of uncertainties regarding effect sizes, confidence intervals around these PAFs are wide. Over 2025-2040, none of the interventions are estimated to have impacts significantly different from zero, due to limited impact on secondary economic outcomes. The authors conclude that although povery is a significant driver of HIV, STIs and teenage pregnancy, precise quantification is challenging. In addition, extrapolating economic strengthening interventions to daily practice is expected to have insufficient impact on socioeconomic status to reduce HIV and STIs significantly at a population level.

The paper addresses a very important topic. I congratulate the authors on a well-designed and performed study. The article is well written. I have one minor comment: please add the national per-capita income in the asbtract as this is an important factor in the analysis.

---

* Please upload any figures associated with your paper as individual TIF or EPS files with 300dpi resolution at resubmission; please read our figure guidelines for more information on our requirements: http://journals.plos.org/plosmedicine/s/figures. While revising your submission, please upload your figure files to the PACE digital diagnostic tool, https://pacev2.apexcovantage.com/. PACE helps ensure that figures meet PLOS requirements. To use PACE, you must first register as a user. Then, login and navigate to the UPLOAD tab, where you will find detailed instructions on how to use the tool. If you encounter any issues or have any questions when using PACE, please email us at PLOSMedicine@plos.org.

* Please ensure that the study is reported according to the appropriate guideline and include the completed checklist as Supporting Information. When completing the checklist, please use section and paragraph numbers, rather than page numbers. Please add the following statement, or similar, to the Methods: "This study is reported as per [XXXX] guideline (S1 Checklist)."

FIGURES AND TABLES

SUPPLEMENTARY MATERIAL

REFERENCES

[STUDY TYPE-SPECIFIC REQUESTS - DELETE SECTIONS AS NECESSARY]

RCTs [REFER TO RCT CHECKLIST AND MEETING NOTES FOR DETAILS TO ADD]

* PLOS Medicine requires that all trials be prospectively registered in one of registries recognized by WHO. Please ensure that study registration details are included in the Methods section.

* Please structure the Methods section using the following sub-headings: Study design and participants, Randomization and masking, Procedures, Outcomes, Statistical analysis.

* The following outcomes measures [ADD DETAILS AS NEEDED OR DELETE BULLET POINT] appear to differ between the submitted manuscript and the protocol [and/or trial registry]. Please clarify and explain all discrepancies between the paper and protocol. If the outcomes were not prespecified in the protocol, please define them in the Methods (Outcomes section) as post hoc and explain why they were added. Post-hoc comparisons should be presented as hypothesis generating rather than conclusive.

* Please ensure that all prespecified outcomes (primary, secondary, and exploratory) are listed in the Methods/Outcomes section and indicate whether there are outcomes that are not presented in the current report.

* Please specify the dates (Month Day, Year) during which study enrollment and follow up occurred.

* Please include absolute numbers wherever you report percentages; eg, n/N (%)

* Please present the safety data for the study including numbers of specific events and whether or not adverse events are thought to be related to treatment. AEs should be reported in the abstract, per CONSORT and CONSORT-Harms.

* Please complete the CONSORT checklist (https://www.equator-network.org/reporting-guidelines/consort/) and ensure that all components of CONSORT are present in the manuscript, including how randomization was performed, allocation concealment, blinding of intervention, definition of lost to follow-up, power statement. When completing the checklist, please use section and paragraph numbers, rather than page numbers.

* Please report your abstract according to CONSORT for abstracts, following the PLOS Medicine abstract structure (Background, Methods and Findings, Conclusions) https://www.equator-network.org/reporting-guidelines/consort-abstracts/

* If your trial had to undergo important modifications in response to extenuating circumstances, please complete the CONSERVE-CONSORT checklist and provide in your Supporting Information; (https://www.equator-network.org/reporting-guidelines/guidelines-for-reporting-trial-protocols-and-completed-trials-modified-due-to-the-covid-19-pandemic-and-other-extenuating-circumstances-the-conserve-2021-statement/). When completing the checklist, please use section and paragraph numbers, rather than page numbers.

* In keeping with our commitment to Open Science, please include the study protocol document and analysis plan (including any amendments) as Supporting Information to be published with the manuscript if accepted.

* Please note that PLOS Medicine requires prospective, public registration of a data sharing plan (as part of mandatory clinical trials registration) for all clinical trials that began enrollment on or after January 1, 2019, in accordance with ICMJE requirements.

OBSERVATIONAL STUDIES

* Abstract: Please include the study design, population and setting, number of participants, years during which the study took place (enrollment and follow up), length of follow up, and main outcome measures.

* Please ensure that the study is reported according to the STROBE (or appropriate STOBE extension) guideline (available from: https://www.equator-network.org/reporting-guidelines/strobe) and include the completed STROBE (or STROBE extension) checklist as Supporting Information. Please add the following statement, or similar, to the Methods: "This study is reported as per the Strengthening the Reporting of Observational Studies in Epidemiology (STROBE) guideline (S1 Checklist)." When completing the checklist, please use section and paragraph numbers, rather than page numbers.

* [FOR POPULATION HEALTH/REGISTRY STUDIES] Please ensure that the study is reported according to the RECORD guideline (available from https://www.record-statement.org) and include the completed checklist as Supporting Information. Please add the following statement, or similar, to the Methods: "This study is reported as per the Reporting of Studies Conducted using Observational Routinely-Collected Data (RECORD) guideline (S1 Checklist)." When completing the checklist, please use section and paragraph numbers, rather than page numbers.

* [FOR POPULATION HEALTH ESTIMATES] Please ensure that the study is reported according to the GATHER statement (available from https://www.equator-network.org/reporting-guidelines/gather-statement) and include the completed checklist as Supporting Information. Please add the following statement, or similar, to the Methods: "This study is reported as per the Guidelines for Accurate and Transparent Health Estimates Reporting (GATHER) statement (S1 Checklist)." When completing the checklist, please use section and paragraph numbers, rather than page numbers.

* [FOR MEDIATION ANALYSES] We recommend that the study is reported according to the AGReMA statement (https://agrema-statement.org/#:~:text=AGReMA%20is%20an%20evidence%2D%20and,randomised%20trials%20and%20observational%20studies) and include the completed checklist as Supporting Information. Please add the following statement, or similar, to the Methods: "This study is reported as per the Guideline for Reporting Mediation Analyses (AGReMA) statement (S1 Checklist)." When completing the checklist, please use section and paragraph numbers, rather than page numbers.

* For all observational studies, in the manuscript text, please indicate: (1) the specific hypotheses you intended to test, (2) the analytical methods by which you planned to test them, (3) the analyses you actually performed, and (4) when reported analyses differ from those that were planned, transparent explanations for differences that affect the reliability of the study's results. If a reported analysis was performed based on an interesting but unanticipated pattern in the data, please be clear that the analysis was data driven.

* Please state in the Methods section whether the study had a prospective protocol or analysis plan. If a prospective analysis plan (from your funding proposal, IRB or other ethics committee submission, study protocol, or other planning document written before analyzing the data) was used in designing the study, please include the relevant document(s) with your revised manuscript as a Supporting Information file to be published alongside your study and cite it in the Methods section. A legend for this file should be included at the end of your manuscript. If no such document exists, please make sure that the Methods section transparently describes when analyses were planned, and when/why any data-driven changes to analyses took place. Changes in the analysis, including those made in response to peer review comments, should be identified as such in the Methods section of the paper, with rationale.

MODELLING STUDIES

The following list is derived from Geoffrey P Garnett, Simon Cousens, Timothy B Hallett, Richard Steketee, Neff Walker. Mathematical models in the evaluation of health programmes. (2011) Lancet DOI:10.1016/S0140-6736(10)61505-X:

* If pertinent, please provide a diagram that shows the model structure, including how the natural history of the disease is represented, the process and determinants of disease acquisition, and how the putative intervention could affect the system.

* Please provide a complete list of model parameters, including clear and precise descriptions of the meaning of each parameter, together with the values or ranges for each, with justification or the primary source cited and important caveats about the use of these values noted.

* Please provide a clear statement about how the model was fitted to the data, including goodness-of-fit measure, the numerical algorithm used, which parameter varied, constraints imposed on parameter values, and starting conditions.

* For uncertainty analyses, please state the sources of uncertainties quantified and not quantified [can include parameter, data, and model structure].

* Please provide sensitivity analyses to identify which parameter values are most important in the model. Uncertainty estimates seek to derive a range of credible results on the basis of an exploration of the range of reasonable parameter values. The choice of method should be presented and justified.

* Please discuss the scientific rationale for the choice of model structure and identify points where this choice could influence conclusions drawn. Please also describe the strength of the scientific basis underlying the key model assumptions.

* For studies that develop a prediction model or evaluate its performance, please ensure that the study is reported according to the TRIPOD statement (https://www.equator-network.org/reporting-guidelines/tripod-statement) and include the completed checklist as Supporting Information. Please add the following statement, or similar, to the Methods: "This study is reported as per the Transparent Reporting of a Multivariable Prediction Model for Individual Prognosis Or Diagnosis (TRIPOD) statement (S1 Checklist)." For studies using machine learning, please use the TRIPOD-AI checklist. When completing the checklist, please use section and paragraph numbers, rather than page numbers.

DIAGNOSTIC STUDIES

* Please ensure that the study is reported according to the STARD guideline (https://www.equator-network.org/reporting-guidelines/stard/) and include the completed STARD checklist as Supporting Information. Please add the following statement, or similar, to the Methods: "This study is reported as per the Standards for Reporting of Diagnostic Accuracy (STARD) guideline (S1 Checklist)." When completing the checklist, please use section and paragraph numbers, rather than page numbers.

* Please structure your Abstract according to STARD for Abstracts (https://www.equator-network.org/reporting-guidelines/stard-abstracts/).

* Please structure the Methods section using the following sub-headings: Study design, Participants, Test methods, Analysis.

* Please include a diagram to describe the flow of participants through the study (typically figure 1).

MENDELIAN RANDOMIZATION STUDIES

* Please ensure that the study is reported according to the STROBE-MR guideline (https://www.equator-network.org/reporting-guidelines/strobe/) and include the completed STROBE-MR checklist as Supporting Information. Please add the following statement, or similar, to the Methods: "This study is reported as per the Strengthening the Reporting of Observational Studies in Epidemiology (STROBE) guideline, specific for mendelian randomization (S1 Checklist)." When completing the checklist, please use section and paragraph numbers, rather than page numbers.

* In the Introduction, please describe the exposure and the evidence for a potential causal relationship between exposure and outcome.

* In the Methods, please explicitly state the 3 core instrumental variable assumptions for the main analysis (relevance, independence, and exclusion restriction), as well assumptions for any additional or sensitivity analysis.

* In the Methods, please describe the MR estimator (e.g., 2-stage least squares, Wald ratio) and related statistics. Detail the included covariates and, in case of 2-sample MR, whether the same covariate set was used for adjustment in the 2 samples.

* If you are presenting an instrumental variable estimate, please compare this to the conventional observational estimate.

* Report the associations between genetic variant and exposure and between genetic variant and outcome, preferably on an interpretable scale.

* Report MR estimates of the relationship between exposure and outcome and the measures of uncertainty from the MR analysis, on an interpretable scale, such as odds ratio or relative risk per SD difference.

* If relevant, please consider translating estimates of relative risk into absolute risk for a meaningful time period.

* Please consider including plots to visualize results (e.g., forest plot, scatterplot of associations between genetic variants and outcome vs between genetic variants and exposure).

SURVEY-BASED STUDIES

* Please ensure that the study is reported according to the CROSS guideline (https://www.equator-network.org/reporting-guidelines/a-consensus-based-checklist-for-reporting-of-survey-studies-cross/) and include the completed CROSS checklist as Supporting Information. Please add the following statement, or similar, to the Methods: "This study is reported as per A Consensus-Based Checklist for Reporting of Survey Studies (CROSS) guideline (S1 Checklist)." When completing the checklist, please use section and paragraph numbers, rather than page numbers.

* Please report your survey response rates according to AAPOR recommendations (https://aapor.org/standards-and-ethics/best-practices/)

* Please define how the population surveyed was sampled.

* Please compare characteristics of respondents and nonrespondents if possible.

* If sequential waves of the survey were sent, please specify whether the characteristics of respondents changed over time or remained constant.

* Please include the survey response rate in the Abstract.

* Please include a copy of the survey in the supplementary files.

SYSTEMATIC REVIEWS & META-ANALYSES

* Please report your SR/MA according to the PRISMA guidelines provided at the EQUATOR site. http://www.equator-network.org/reporting-guidelines/prisma/. Please provide the completed PRISMA checklist as Supporting Information. When completing the checklist, please use section and paragraph numbers, rather than page numbers. Please add the following statement, or similar, to the Methods: "This study is reported as per the Preferred Reporting Items for Systematic Reviews and Meta-Analyses (PRISMA) guideline (S1 Checklist)."

* Abstract: Please report your abstract according to PRISMA for abstracts (https://doi.org/10.1371/journal.pmed.1001419) following the PLOS Medicine abstract structure (Background, Methods and Findings, Conclusions). Please ensure you provide dates of search, data sources, number of studies included, types of study designs included, eligibility criteria, and synthesis/appraisal methods.

* Please note that we expect searches to be updated to within 6 months of the time of submission.

QUALITATIVE STUDIES

* Please report your qualitative study according to the appropriate study design provided at (http://www.equator-network.org/?post_type=eq_guidelines&eq_guidelines_study_design=qualitative-research&eq_guidelines_clinical_specialty=0&eq_guidelines_report_section=0&s=) and provide the relevant completed checklist as a supplemental file. In the checklist, please include sufficient text excerpted from the manuscript to explain how you accomplished all applicable items. When completing checklists, please use section and paragraph numbers, rather than page numbers.

* We recommend that authors use the COREQ checklist, or other relevant checklists listed by the Equator Network, such as the SRQR, to ensure complete reporting (see: http://www.equator-network.org/?post_type=eq_guidelines&eq_guidelines_study_design=qualitative-research&eq_guidelines_clinical_specialty=0&eq_guidelines_report_section=0&s=). Please add the following statement, or similar, to the Methods: "This study is reported as per the Consolidated criteria for reporting qualitative research (COREQ): a 32-item checklist for interviews and focus groups (S1 Checklist)."

* In general, we expect qualitative studies to include the following: 1) defined objectives or research questions; 2) description of the sampling strategy, including rationale for the recruitment method, participant inclusion/exclusion criteria and the number of participants recruited; 3) detailed reporting of the data collection procedures; 4) data analysis procedures described in sufficient detail to enable replication; 5) a discussion of potential sources of bias; and 6) a discussion of limitations.

HEALTH ECONOMICS / COST-EFFECTIVENESS STUDIES

* Please ensure that the study is reported according to the CHEERS guideline (available from: https://www.equator-network.org/reporting-guidelines/cheers) and include the completed checklist as Supporting Information. Please add the following statement, or similar, to the Methods: "This study is reported as per the Strengthening the Consolidated Health Economic Evaluation Reporting Standards 2022 (CHEERS 2022) Statement (S1 Checklist)." When completing the checklist, please use section and paragraph numbers, rather than page numbers.

---

## [Decision Letter · Decision Letter 2]

21 Oct 2025

Dear Dr. Johnson,

Thank you very much for re-submitting your manuscript "Money, jobs or schooling? A model-based evaluation of economic strengthening in South Africa and its impact on HIV, sexually transmitted infections and teenage births" (PMEDICINE-D-25-01888R2) for review by PLOS Medicine.

I have discussed the paper with my colleagues and the academic editor and it was also seen again by two of the original reviewers. I am pleased to say that provided the remaining editorial, reviewer and production issues are dealt with, we are planning to accept the paper for publication in the journal.

Please note that reviewers 1 and 2 require additional discussion of the study conclusions and limitations, additional reporting for full transparency and reproducibility, further discussion of the rationale for the model and approach, as well as some sensitivity analyses.

[LINK]

We look forward to receiving the revised manuscript by Oct 28 2025 11:59PM.   

Sincerely,

Alison

Alison Farrell, Ph.D.

Senior Editor 

PLOS Medicine

plosmedicine.org

Requests from Editors:

* Please confirm that your title complies with PLOS Medicine's style. Your title must be nondeclarative and not a question. It should begin with main concept if possible. "Effect of" should be used only if causality can be inferred, i.e., for an RCT. Please place the study design ("A randomized controlled trial," "A retrospective study," "A modelling study," etc.) in the subtitle (ie, after a colon). Please consider: Evaluation of economic strengthening in South Africa and its impact on HIV, sexually transmitted infections and teenage births: a modeling study

* Please confirm that your abstract complies with our requirements, including format (three sections: Background, Methods and Findings, and Conclusions) and providing all the information relevant to this study type https://journals.plos.org/plosmedicine/s/submission-guidelines#loc-abstract

* Please ensure that the Introduction ends with a clear description of the study question or hypothesis.

* Please ensure that all abbreviations are defined at first use throughout the text.

* Please confirm that all numbers presented in the abstract are present and identical to numbers presented in the main manuscript text.

Line 49 of Background: Qualify ‘these’. Incidences?events?

Line 53: The first sentence should explain what you did in this study. Please rephrase to indicate what the model was used to assess (reword “to reflect effects of”).

Line 54: remove comma after Africa

Please use active voice throughout. E.g. “We estimated…”

Please use commas rather than hyphens in confidence intervals.

Lines 65-66, impact used twice. Please clarify what the first ‘impact’ relates to (which outcomes).

Line 66 and lines 109-110: please rephrase “would come close”. Please be precise in how you frame the reduction in teenage births. Please also do not repeat phrasing of the Abstract in the Author Summary.

Please add an additional qualifier to the last sentence of the Abstract to explain what the recently trialled econonomic interventions aimed to do.

In the final bullet point of ‘What Do These Findings Mean?’, please include the main limitations of the study in non-technical language.

In the main text, Background should be called Introduction.

Line 191: qualify ‘impactful’ (what do they impact?).

Please provide a 1-2 sentence introduction to the Results section.

Please clarify line 468 for the general reader.

The figure legend to Figure 2 should not start with definitions of acronyms. Please revise start with an explanation of what is in the figure. Each figure panel should be described separately.

* In the abstract, please include the important dependent variables that are adjusted for in the analyses.

* Where data points are discrete, please ensure that they are depicted in the figures as discrete data and not as a continuous line.

* Please provide the unadjusted comparisons as well as the adjusted comparisons in all relevant Tables.

Modeling studies should include the following components, as derived from Geoffrey P Garnett, Simon Cousens, Timothy B Hallett, Richard Steketee, Neff Walker. Mathematical models in the evaluation of health programmes. (2011) Lancet DOI:10.1016/S0140-6736(10)61505-X:

* Please provide a diagram that shows the model structure, including how the disease natural history is represented, the process and determinants of disease acquisition, and how the putative intervention could affect the system.

* Please provide a complete list of model parameters, including clear and precise descriptions of the meaning of each parameter, together with the values or ranges for each, with justification or the primary source cited, and important caveats about the use of these values noted.

* Please provide a clear statement about how the model was fitted to the data including where relevant goodness-of-fit measure, the numerical algorithm used, which parameter varied, constraints imposed on parameter values, and starting conditions.

* For uncertainty analyses, please state the sources of uncertainties quantified and not quantified this can include parameter, data, and model structure.

* Please provide sensitivity analyses to identify which parameter values are most important in the model. Uncertainty estimates seek to derive a range of credible results on the basis of an exploration of the range of reasonable parameter values. The choice of method should be presented and justified.

* Please discuss the scientific rationale for this choice of model structure and identify points where this choice could influence conclusions drawn. Please also describe the strength of the scientific basis underlying the key model assumptions.

Comments from Reviewers:

Reviewer #1: I appreciate the authors' thoughtful and constructive revisions, which have notably improved the transparency and methodological rigour of the manuscript. The majority of my previous concerns—pertaining to the model structure, calibration methodology, justification of assumptions, and the realism of intervention scenarios—have been adequately addressed. The revised version presents clearer descriptions of the model submodules, a more detailed explanation of the calibration procedures and prior specification, and an expanded discussion on the limitations related to generalisability and implementation feasibility.

While the manuscript is now suitable for publication, I offer the following final suggestions that may enhance its robustness and credibility:

1. Consider reporting basic model fit metrics (e.g., root mean square error, mean absolute error) or presenting posterior predictive checks for key outcomes to more clearly demonstrate the model's internal and external validity beyond qualitative assessments.

2. Given the wide credible intervals observed for several outcomes, it would be helpful to provide a more explicit narrative interpretation of these intervals and their implications for uncertainty in policy recommendations.

3. While parameter uncertainty is well-addressed, the manuscript would benefit from additional sensitivity analysis—or at least discussion—regarding structural assumptions, particularly the simplified causal pathways linking socioeconomic status (SES) to health-related behaviours.

4. The model currently assumes immediate behavioural effects following changes in SES. Future iterations could improve realism by incorporating lagged or cumulative effects, especially for longer-term interventions such as educational and vocational support.

Reviewer #2: Most of the comments from the previous reviewers were well addressed, but I still have the remaining concerns:

* Behavioral Mechanism Precision (Reviewer #1, point b): The authors provide detailed explanations of behavioral mechanisms in the supplementary materials and seem hesitant to duplicate this information in the main text due to length concerns. The reviewer may still feel this area could benefit from more inclusion in the main body of the paper.

* Implementation of Interventions (Reviewer #1, point g): The authors acknowledge that the assumptions about intervention implementation are optimistic but justify it as a "best-case scenario" assessment. The reviewer might still prefer a more realistic scaling scenario, even if briefly explored.

* Model structure and scientific rationale (Reviewer #2, point 3): The authors added a few sentences in the Introduction to make the rationale clearer. "They [mathematical models] also play an important role in translating evidence from randomized trials, which produce short-term estimates of impact at an individual or community level, into estimates of likely longer-term impact at a population level.... Agent-based models, which represent individual-level variation in health risk exposure and outcomes, are well suited to representing complex causal pathways and diverse outcomes, and are increasingly utilized in assessing socio-economic disparities in health [28]." This response only addresses why modeling is important, but did not mention the rationale for the model structure.

* The reasons for this lack of impact should be explored in more detail (Reviewer #2, point 14): The authors showed in Figure 3 that the interventions have only a modest impact on socioeconomic indicators, and we have added further detail to Table 4 showing that the interventions have almost zero impact on intermediate behavioural outcomes. "These modest effects on socioeconomic indicators are further diluted at each subsequent step in the causal pathway, so that the final effects on reproductive health outcomes are negligible." However, the reasons for this lack of impact is still not clear.

* Number of Simulations (Reviewer #1, point f): The authors doubled the posterior sample size and the results have been updated. But authors acknowledge that this only partially addresses the reviewer's concern, they have mentioned constraints of computing power.

* Accounting for Generalizability of Trial Populations (Reviewer #1, point j): The authors state that the focus is on the heterogeneity in effects across settings and they address this by including a random effect term in the likelihood function. However, this cannot address the problem of generalizability. Instead, the authors would test the difference between the trial populations and the generalized population, and generalize those through modeling using through g-formula or inverse probability weighting.

[LINK]

---

## [Editor Report · Decision Letter 3]

11 Nov 2025

Dear Dr Johnson, 

On behalf of my colleagues and the Academic Editor, Aaloke Mody, I am pleased to inform you that we have agreed to publish your manuscript "Evaluation of economic strengthening in South Africa and its impact on HIV, sexually transmitted infections and teenage births: a modelling study" (PMEDICINE-D-25-01888R3) in PLOS Medicine.

**Please include an Acknowledgements section in your manuscript.

PRESS

Sincerely, 

Alison Farrell, Ph.D. 

Senior Editor 

PLOS Medicine